

# Benchmarking soil multifunctionality

E.R. Jasper Wubs[1]

[1]Department Terrestrial Ecology, Netherlands Institute of Ecology (NIOO-KNAW), P.O. Box 50, 6700 AB, Wageningen, The Netherlands

*Correspondence to*: E.R. Jasper Wubs (j.wubs@nioo.knaw.nl)

**Abstract.** Healthy soils provide multiple functions that importantly contribute to human wellbeing, including primary production, climate and water regulation, and supporting biodiversity. These functions can partially be combined and some functions also clearly trade-off: this motivates soil multifunctionality research. Society needs scientists to help assess which soils are best for which soil functions and to determine appropriate long-term management of any given soil for optimal

function delivery. However, for both tasks science lacks coherent tools and in this paper I propose a way forward.

Critically, we lack a common measurement framework that pins soil functioning measurements on a common scale. Currently the field is divided with respect to the methods we use to measure and assess soil functioning and indicators thereof. Only three indicator variables (SOM, acidity, and available P) were commonly measured (>70% of schemes) across 65 schemes that aim to measure soil health or quality, and no biological measure is implemented in more than 30% of the

65 schemes. This status quo prevents us from systematically comparing across and within soils; we lack a soil multifunctionality benchmark.

We can address this limitations systematically by setting a common measurement system. To do this, I propose to use latent variable modelling based on a common set of functional measurements, to develop a common 'IQ test for soils'. I treat soil functions as latent variables, because they are complex processes that cannot be measured directly, we can only

detect drivers and consequences of these complex processes. Latent variable modelling has a long history in social, economic and psychometric fields, where it is known as factor analysis. Factor analysis aims to derive common descriptors – the factors – of hypothesized constructs by linking measurable response variables together on a common scale.

Here, I explain why such a new approach to soil multifunctionality and soil health is needed and how it can be operationalized. The framework developed here is only an initial proposal, the issue of soil multifunctionality is too complex

and too important to be addressed in one go. It needs to be resolved iteratively by bands of scientist working intensively together. We need to bring our best science together, in a collaborative effort, to develop progressively more refined ways of sustainably managing one of humanity's most precious resources: our soils.



## 1 Introduction

Human actions are perturbing the Earth system beyond its planetary boundaries, particularly for biodiversity, climate and

flows of phosphorus and nitrogen, while we also need to provide sustainable social livelihoods across the globe (Fanning et

al., 2022; Lade et al., 2020; Steffen et al., 2015). Agricultural production is a main driver of environmental problems, due

land use change, depletion of freshwater resources, and pollution of aquatic and terrestrial ecosystems (Springmann et al.,

2018). In addition, modern agriculture will have to adapt to global limits on mineral phosphorus supply (Blackwell et al.,

2019) and increasing regulation of pesticide use (Tang and Maggi, 2021). This means land-bound agriculture will have to

increasingly rely on the internal functional capacity of soils, e.g. to recycle nutrients and supress diseases, and thus soil health.

Likewise, regulation of the climate, through carbon sequestration and reducing greenhouse gas emissions (Lehmann et al.,

2020), and the provision of habitat for aboveground biodiversity, to bend the curve of biodiversity loss (Leclère et al., 2020),

are directly and indirectly linked to soil health. Furthermore, soil biodiversity importantly contributes to climate change

adaptation, by storing precipitation in soils (Lal, 2020), and achieving ONE Health through removal of contaminants and

preventing disease spread (Wall et al., 2015). Indeed, soil and soil health are at the heart of achieving many of the UN

Sustainable Development Goals for 2030 (Keesstra et al., 2016; Lal et al., 2021) and the European Green Deal (Montanarella

and Panagos, 2021).

Soil health, defined here as 'the continued capacity of soils to deliver the multiple soil functions on which society depend',

takes centre stage in policy and practise with respect to soils worldwide (Van der Putten et al., 2023; Veerman et al.,

2020). However, currently the field is divided with respect to the methods we use to measure and assess soil functioning

and indicators thereof. Only three indicator variables (SOM, acidity, and available P) were commonly measured (>70% of

schemes) across 65 schemes that aim to measure soil health or quality, and no biological measure is implemented in more

than 30% of the 65 schemes (Bünemann et al., 2018). Indeed, until very recently there was no national or European level

monitoring system that could address the key functions of soils comprehensively (Creamer et al., 2022; Van Leeuwen et

al., 2017), although steps in this direction are now being taken (Norris et al., 2020; Orgiazzi et al., 2022; Zwetsloot et

al., 2021). It is clear that further harmonization in methods and quantification is urgently needed.

Partly, I think this plethora of methods and approaches stems from an oversimplified, often correlational understanding of

the causal linkages driving soil multifunctionality, equipment availability in laboratories involved, and a 20+ year policy

pressure to deliver easy to implement indicators (Creamer et al., 2022), which prevented the zooming-out needed to better

understand the soil systematically (Harris et al., 2022). Indeed, what we need are: "new analytical and conceptual approaches

[…] that capture systems characteristics of soil health, in order to operationalize both monitoring soil health itself and



understanding soil-health effects on soil functions" (Lehmann et al., 2020). However, systemic perspectives that integrate

soil functions and responses are in their infancy (Vogel et al., 2018). It is unclear how to manage the soil functions (Baveye

et al., 2016), and how to link functions to soil processes (Vogel et al., 2018), but see (Creamer et al., 2022; Vogel et al.,

2018). Integrating all soil processes is highly complex, because soil properties are spatially heterogenous and the interactions

in soil are typically non-linear (Vogel et al., 2018).  Soil biology is a key missing ingredient, but its complexity is paralyzing the

soil health literature (Creamer et al., 2022; Lehmann et al., 2020; Van Leeuwen et al., 2017). We know that soil

biodiversity drives soil multifunctionality (Delgado-Baquerizo et al., 2016; Wagg et al., 2014), but the causal relation to

soil functioning for many organisms is not clear (Creamer et al., 2022). Many soil microbial variables measured are hard to

interpret and are insufficiently benchmarked to allow inferences about soil health (Fierer et al., 2021). Furthermore, most

research focuses on soil health in an agricultural context (Debeljak et al., 2019; Fierer et al., 2021), but we also need to

understand and quantify it in forestry, nature management, drinking water production areas, industrial and urban areas,

which are strongly underrepresented (Norris et al., 2020; Orgiazzi et al., 2022).

To move forward, we first need to know what kind of information society needs from soil science. In this context I think the

main research tasks are:

1.  Determine which soils are best for which function (FAO and ITPS, 2015), and which functions can be combined

            (synergies) and which cannot (trade-offs),

        2.  Determine the functional shape of the interrelations among soil functions and the governing mechanisms.

        3.  Determine the mechanistic drivers of the multiple functions of soils over a long-term perspective.

        4.  Determine how multifunctionality of individual soils can be optimized.

5.  Develop a simple and effective indicator set to monitor status and trends of soil functions and multifunctionality.

When we know these, we can start the spatial optimization of multifunctional soil use (van Wijnen et al., 2012), and if we

understand the long-term impacts and dynamics with respect to the functions and their drivers we can do so for long-term

sustainable use.

To do these tasks well, we need to get organized as a scientific community. We need to come up with a model of the

interrelations among the soil functions and their drivers, and we need to set a common measurement system for the multiple

functions of soil. We need a balanced set of indicators, that reflect soil biology, chemistry and physics, but that are geared

towards soil functioning (Lehmann et al., 2020). So far, selection of soil biological indicators was driven by well-known

methods, feasibility in general laboratories and costs, but they should be based on sound understanding of how the indicators

link to soil functioning mechanistically (Creamer et al., 2022; Lehmann et al., 2020; Vogel et al., 2018). New proposals





typically try to go from soil processes to functions in one go, but soil is complex (Young and Crawford, 2004) and so far this approach has been defeated by this complexity. In many cases, the drivers of soil functions, either direct or indirect, are used implicitly or explicitly as proxies for the functions themselves. For example, soil nutrient content is used as a proxy for soil fertility (Daou and Shipley, 2019), or microbial biomass as a proxy for carbon storage (Wiesmeier et al., 2019), which in both

cases do contribute to the function, but are not nearly a complete description of it. We can make steps forward by formally separating the causes and consequences, the predictors and the indicators, of soil functioning and by linking them to the underlying processes and environmental and management context. I propose that we can do so by applying latent variable models and structural causal modelling to soil multifunctionality research.

My aim with this paper is to propose a new methodology for measuring soil functioning and soil multifunctionality. It is based on the well-established technique of latent variable modelling commonly used in psychometry, economics and the social sciences at large. The next step after this will be to develop a causal model of how trade-offs and synergies among soil functions are mechanistically regulated. If we define soil health as the continued capacity of soils to deliver the multiple soil functions on which society depend, then what are soil functions? Here, I define soil functions as soil processes, physical,

chemical, or biological in nature, acting singly or in combination. These functions can be beneficial for human society, but can also be involved in the internal functioning of ecosystems, without direct human benefits, i.e. soil functioning for the sake of the ecosystem itself. For consistency, perhaps 'soil functions for human wellbeing' should perhaps be called 'soil services', as a specific form of ecosystem services.

## 2 Conceptual approach to soil multifunctionality

Great mathematical frameworks now exist to combine multiple functions into one aggregate measure of multifunctionality (Byrnes et al., 2014, 2023), and they could be used to signal that 'something is wrong' with soil functioning. However, understanding which soils perform all functions best in aggregate, e.g. the highest average soil function, is not informative enough to guide sustainable use of soils (Bradford et al., 2014; Lehmann et al., 2020). We need to know which soils perform which functions well, and to what extend the functions can be combined in a single soil. So instead of focussing on univariate

summary statistics of multifunctionality, we need to come up with a multivariate, but still simple and communicable, representation for soil multifunctionality (Lehmann et al., 2020; Zwetsloot et al., 2021). Multivariate models of multifunctionality have been developed, including network approaches that, I think, can be valuable in exploratory investigations (Siwicka et al., 2021). Others developed elegant multivariate models to estimate the influence of different drivers on functions and interrelations among functions (Dooley et al., 2015). However, all these approaches are correlational

in nature, leaving the causal relationships that induced these correlations potentially unexamined (Shipley, 2016). I think this





is problematic, because of 1) potential paradoxes in the data that no amount of big data can resolve (e.g. Simpsons paradox) and 2) difficulties in generalizing the results of analyses to other contexts. Posing a mechanistic model that links soil functions *a priori*, which is iteratively improved in the face of new data, can resolve both of these issues. In addition, hypothesizing such mechanistic models will help in stabilizing the set of measured 'functions' now rampant in the literature, by excluding

those indicators that are actually stocks or ecosystem properties and not processes (Garland et al., 2021; Lehmann et al., 2020). Confronting the hypothesized models with data and proposing improvements can be done with structural equations modelling (Box 1). But, how to organize the complexity of soils and soil functioning in one model?

**Box 1. Causal inference, structural equation modelling, and latent variables – a short introduction**

"Correlation is not causation" is a central piece of endemic wisdom we scientists throw at one another on a regular basis. However, its complement "causation implies correlation" is much less known, due to Karl Pearson's (Pearson, 1911) crusade on causality. Nevertheless, it is the central concept in modern causal analysis (Pearl, 2009; Shipley, 2016). The modern causal revolution arose from the pioneering work of

population geneticist Sewall Wright, who developed path analysis (Wright, 1921, 1934), a method to estimate causal effects from observational data. His method was ignored by statisticians and biologists for decades, because it did not fit with the views of the dominant schools of statistics headed by Karl Pearson and Ronald A. Fisher (Shipley, 2016). Instead, the method was refined within economics, sociology, political

science and psychology (e.g. Jöreskog, 1967).

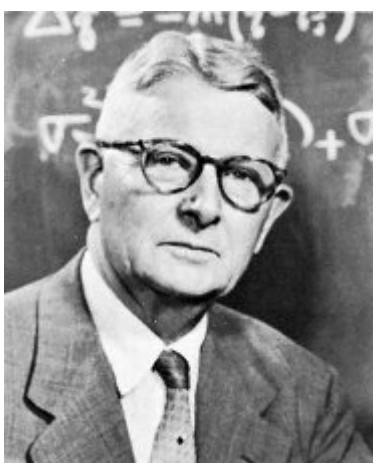

**Sewall Wright FRS (1889-1988)**

(Source Wikipedia)

Path analysis was transformed into structural equations modelling (SEM), which uses maximum likelihood (ML) estimation to test causal multivariate hypotheses. The multivariate hypotheses are specified as a graph, specifically a directed acyclical graph, which captures the hypothesized causal relationships among the variables

involved. The central idea is beautifully simple: if the specified causal hypothesis is true then we can predict which variables should be correlated and which not, the latter are considered to be conditionally independent. In fact, the method depends on predicting the covariance matrix of the variables, comparing it to the observed covariance matrix and testing the model fit (using an ML $\chi^2$ test). If the model does not fit the data (e.g. $\chi^2$ p < 0.05) then the hypothesized causal graph is rejected. If there is no lack of fit, then one concludes that the data are consistent with the causal processes hypothesized (until in the

next paper someone else shows you wrong, of course). For SEM to work it needs to assume linear relationships and multivariate normal distributions of the variables involved, but it comes with the major advantage that it can estimate latent



variables. Latent variables (LVs) are variables that were not measured or even cannot be measured. LVs are a way to quantify the unmeasurable!

LVs are extremely important concepts, as many things cannot be measured (Shipley, 2016). For instance, we cannot measure air temperature, which is the average kinetic energy of the molecules in the air, we can only measure its effects on e.g. the expansion of mercury in a capillary column (a mercury thermometer), or the change in electrical voltage in a thermocouple. These observed variables are of course causally linked to the latent quantity temperature, but they are observed with measurement error. Misspecifying this dependence relation in a causal model, thus conflating air temperature ('heat') with
the readings of your thermometer (translated to °C), can lead to an erroneous test of the causal model, because it leads to a different expected covariance structure and thus different conditional independence claims. Latent variable models (LVMs) are a way to get around this problem, by specifying that the observed variable (thermocouple voltage) is caused by the quantity of interest (air temperature), but it is observed with error and therefore correlated, but not identical. This situation is treated by 'measurement models' (Fig. 1), a subsection of LVM developed in the social sciences. To parameterize and test
a single LV, four indicator variables need to be measured to have sufficient degrees of freedom, although this can be relaxed if the model entails multiple causally related LVs. LVs are also used to represent more hypothetical variables, e.g. concepts such as genes, atoms and intelligence are examples of latent variables. These examples are successful latent concepts, there are also problematic ones, such as 'ether'. Choosing, developing and justifying latent variables is, perhaps, the most difficult aspect of structural equation modelling.


Recently, the SEM toolbox was expanded with a new estimation and testing method based on d-seperation. D-seperation is a criterion used to derive conditional independence claims, specifying which variables should not be correlated given the *a priori* specified causal model (Shipley, 2000, 2016). The d-separation based approach is flexible and can fully accommodate non-normal data, non-linear functional relationships and nested sampling structures, as it works not with the whole
covariance matrix, but instead it looks at each d-seperation independence claim separately (using partial correlations in its most simple form) and combines this to test the whole causal model using a Fisher's (oh irony) exact C-test (C for combined, the d-sep test; Shipley, 2000). The logic is the same as for ML-based SEM. Given an *a priori* causal model one tests for the conditional independence of variables predicted by the model. Interestingly, the LVM and d-sep approaches can be combined within a single model, if one parameterizes the LVs using ML methods, but performs the statistical testing of the model using
the d-sep approach.

Note, the methods of SEM and LVM are implemented mathematically as regression models, but it is important to realize that the interpretation of SEM is much stronger than ordinary regression models. Ordinary regression models are simple tools



aiming only to predict the effect of X on Y. The goal is prediction, not primarily understanding, although the latter is often attempted. Causal interpretation of regression models is problematic, because parameter estimates and significance depend strongly on the included variables and even their order. In fact, misrepresenting the underlying causal structure can easily lead to entirely the wrong qualitative conclusions, e.g. in the situation called Simpson's paradox (see Supplementary Code), which no amount of data will resolve correctly. SEM, however, is different. It is different not because of its mathematics, it is different because it relies on an *a priori* causal hypothesis to be tested with data. The *a priori* is crucial, when SEM software is used to find the 'best' fitting model by means of model selection tools (e.g. AIC), then Wrights philosophy falls apart and SEM becomes just another regression tool, only to be used for explorative data analysis and hypothesis generation. So, as an analyst using SEM, you get one, and only one, epistemologically sound shot at testing your causal hypothesis. So better think very well about you're *a priori* model! Of course, upon arduously collecting data and then rejecting your model, there is immense temptation to update the model by including new, not *a priori* specified, causal relationships and presenting the updated model in the resultant paper as if it were the original *a priori* model. This is *a posteriori* discovery and again only suitable for exploration and hypothesis generation, not for direct causal interpretation. Therefore, I am strongly in favour of implementing a strict requirement that SEM use for causal hypothesis testing is preceded by the publication of the *a priori* model in a curated, time-stamped, repository. Any updates to the model should be fully reported in the paper, because newly discovered links requiring further testing. In this way, our causal models can be transparently developed and updated. For both SEM and LVM excellent textbooks, reviews and manuals exist (see Grace, 2006; Grace et al., 2010, 2012; Shipley, 2016), as well as for other tools in the causal analysis toolkit (Pearl, 2009). ML fits of LVMs and d-sep tests of SEMs can be obtained in the R packages *lavaan* (Rosseel, 2012) and *piecewiseSEM* (Lefcheck, 2016), respectively. This summary is a condensed version of key points in Shipley (2016).

**/End of Box 1.**

Before we can model interacting soil functions mechanistically, we need a common framework to measure them. For this we have to move beyond using simple indicators, since the processes driving the different functions of soils are complex. Soil fertility, for instance, is a complex soil property that drives the process of primary production. It is complex because many factors contribute to it (Daou and Shipley, 2019) and it changes through time. Higher nutrient availability, but also water content, soil texture and structure interact to shape how well plants grow in a soil. Furthermore, plant species and cultivars respond differently to the different drivers of soil fertility, e.g. some prefer nitrate over ammonium, others are salt or drought tolerant, some can puncture compacted soils and other species not (Grime, 2001). So while it is well possible to build a soil fertility model for individual crops, by accounting for their limiting factors for growth and estimating the functional



relationships to these factors, this is much more difficult to quantify in general with predictive value for all plant and crop species simultaneously (Daou and Shipley, 2019).

Nevertheless, we can borrow the data analytic machinery used in the social sciences to estimate these complex soil traits. In
psychology, economy and other social disciplines, complex properties are measured using latent variable models, and specifically a subsection called 'measurement models', that allow an analyst to infer the status of the complex property by modelling the responses that the property induces (Fig. 1). A well know example is the IQ test that aims to quantify the complex and hard to measure trait intelligence (Spearman, 1904). It does this by fitting a measurement model to the measurable outcomes of intelligence, namely a person's ability to solve particular puzzles in a limited time. Daou & Shipley
have successfully adapted this methodology for quantifying generalized soil fertility (Daou et al., 2021; Daou and Shipley, 2019, 2020), and I propose that we expand their framework to include all major functions of soil, so we can study soil multifunctionality more systematically, I propose an IQ-test for soils.

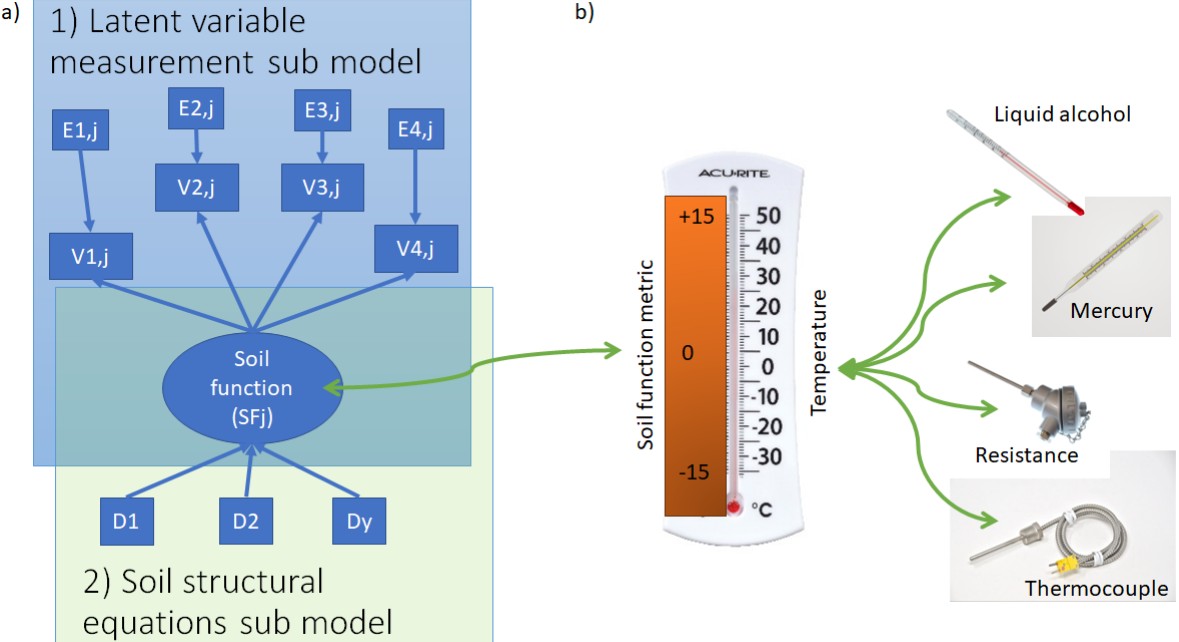

**Figure 1: The two parts of the full soil functioning model.**

a) The two parts of the full soil functioning model including drivers (D1-Dy) and response variables (V1,j-V4,j) and the latent variable representing a single soil function (SFj). See Box 1 for an introduction to structural equation modelling and latent variable modelling. Part one concerns the latent variable measurement sub model involving i indicators measured on each of j soils for each soil function (SF). For example, in the case of primary production the indicators are the growth responses



(RGRij) of four different species used to estimate values for the latent variable generalized soil fertility (FGj). The e's represent

mutually independent measurement errors. See Supplementary Information for an implementation of the model on Dutch

soil samples. Part two concerns the structural equation sub model. It consists in specifying the causal structure linking the y

soil and non-soil variables, drivers (D1 to Dy), that cause SF. For soil fertility, for example, this could be NO3 concentration,

water holding capacity and compaction. b) Analogy of the soil function metrics to quantifying temperature of a water body

as a latent variable using four differently operating thermometers. The latent temperature is estimated using a measurement

model based on readings from a liquid alcohol and a mercury thermometer, based on column height measurements, a

resistance thermometer, which responds to temperature by a change in electrical resistance, and a thermocouple, which

responds to temperature by a change in electrical voltage. By combining these different measurements a more accurate

picture for temperature can be generated. This figure and the example are adapted from Daou and Shipley (2019).

**3 Selecting soil functions and boundary conditions**

Following the Functional Land Management (FLM) framework (Debeljak et al., 2019; Schulte et al., 2014; Zwetsloot et

al., 2021), I focus on four main soil functions of direct importance to society (Fig. 2). The IQ-test for soils will focus on the

soil functions: 1) primary production, driven by soil fertility, 2) climate regulation, consisting of carbon storage and reducing

greenhouse gas emissions (or net GHG consumption by soil), 3) water regulation, composed of water storage and purification

of contaminants, and finally 4) provision of habitat for biodiversity, focussing initially on plant diversity. See proposals for

expansion to other species groups in the discussion. I exclude nutrient cycling, that is included in the FLM, because I think it

is not a soil function beneficial to society in and of itself. Instead, I see it as a structuring principle, nutrient cycling determines

where nutrients are 'invested' and thus which functions 'thrive' (see also Schröder *et al.* 2016). In that sense it is 'the one

ring that rules them all'. Additionally, direct issues with nutrients for society, e.g. low soil fertility and nitrate leaching, are

captured under the other soil functions, respectively primary production and water purification in these examples.



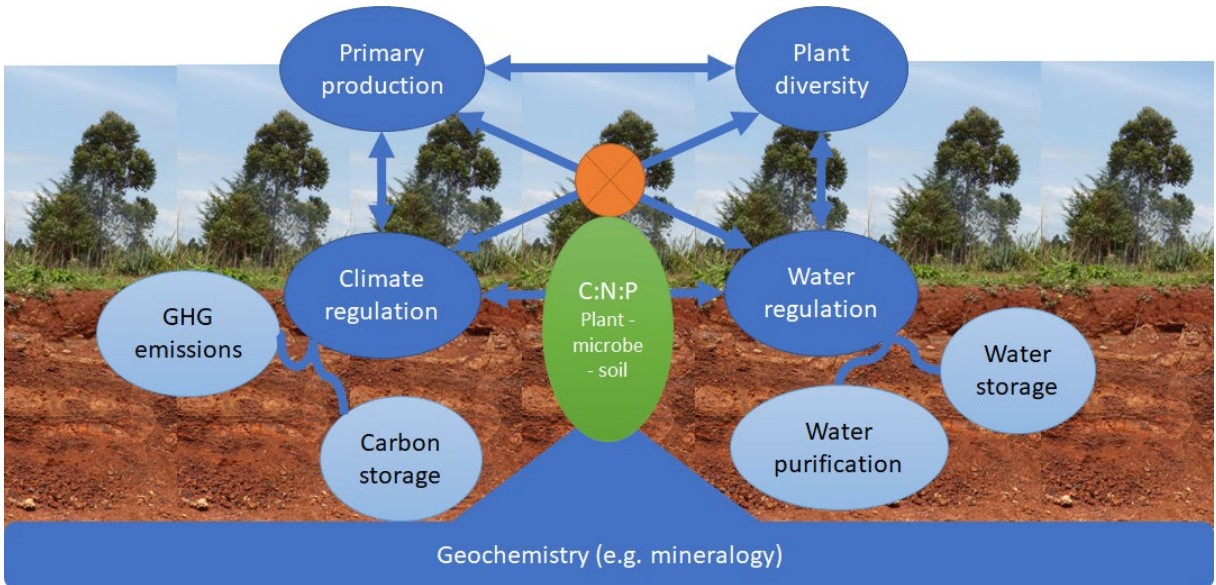

**Figure 2: Soils support human wellbeing in four main areas* (blue circles).**

Climate and water regulation are, respectively, further divided into the carbon storage and reducing greenhouse gas (GHG)
emissions subfunctions, and water storage and purification subfunctions (light blue circles), because of the very different
causal mechanisms in play. The four soil functions are all interrelated, some trading-off, others acting in synergy, because
they all depend on the same basic resources (nutrients, energy, water). I hypothesize that the soil's plant-microbe-soil
stoichiometry determines which functions are preferentially expressed by any given soil. How this regulation plays out is
conditional on the geochemistry of the soil, mainly its mineralogy. Measuring the functions on a common scale and studying
their interrelations using a common causal framework will help us determine how to manage soils for optimal
multifunctionality. * Here I exclude direct and indirect contributions to human health.

The FLM framework was originally designed to integrate over relatively large spatial scales (Schreefel et al., 2022; Schulte
et al., 2014) and uses decision trees, partly based on expert judgement, to generate assessments of the different soil
functions on a semi-quantitative scale (Low-medium-high, Soil Navigator DSS; Debeljak et al., 2019). In addition, the
assessment of different functions is partly based on the same information (Zwetsloot et al., 2021), e.g. SOM is a component
in four out of five functions. How those pre-specified modelling relations affect the observed trade-offs and synergies among
functions is unclear. While I think the efforts made using FLM and the Soil Navigator DSS have great value for society in
recommending changes based on the best knowledge today, I also believe we need to deepen our mechanistic understanding
of the interrelations of the soil functioning and how they can be optimized. For this, I propose we need a measurement and

modelling framework that 1) allows quantitative assessment of soil functions, based on independent data, and 2) assesses functions and drivers at small spatial and temporal resolution (Bradford et al., 2016, 2017; Fierer et al., 2021).

Many processes in soil depend on factors external to soil, such as temperature and water inputs. This contributes to the challenge in using many biological soil health indicators (Fierer et al., 2021), as they can become highly variable in time and space. To get around that, it was proposed to incubate soils under standard conditions, so that only factors internal to a soil would contribute to the observed functioning (Daou and Shipley, 2019). This is the approach I take here as well, and as such the proposed measurement system is focussed on estimating potential soil functioning and multifunctionality, under as set

of soil-external conditions optimal for plant growth. Below, I provide suggestions on how to link these measures to actual *in-situ* rates of soil functioning. Nevertheless, I think this focus on the intrinsic – even though not time invariant – potential soil functioning is important, as it can give the method predictive value for expected *in-situ* soil functioning irrespective of the weather conditions that materialize during the growing season.

## 4 The IQ test for soils - a proposal

Here, I outline a proposal for a standardized soil multifunctionality assay that addresses the key soil functions in the functional land management framework (Schulte et al., 2014). The method is based on incubations of intact soil cores, subjected to several treatments, and measuring responses that are indicative of the underlying soil functions (Fig. 3; Table 1). The methods assume that all soils are sampled in the same way and incubated under standardized conditions, including temperature, light, watering regime, and air humidity, to ensure comparability (see Table 2 for a proposal). The goal is to

estimate the intrinsic capacity of each soil for performing each soil function.



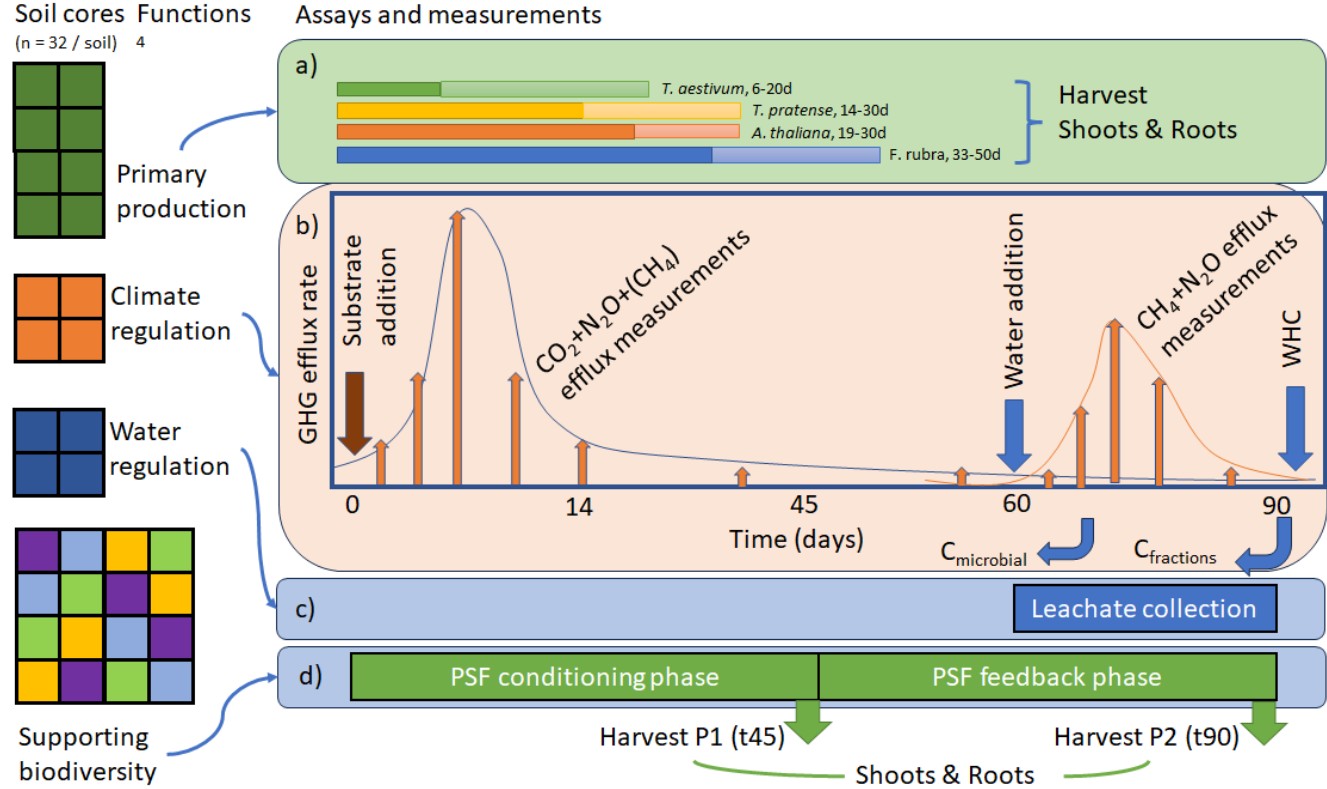

**Figure 3: Design diagram of the soil function measurement setup, version 0.1.**

A soil sampling team will collect 32 soil monoliths (60 mm x 25 cm deep, ~22.6 L soil) per soil. The monoliths are used to

quantify primary production (a) 8 green monoliths, 2 per bio-assay plant species, climate regulation (b) 4 orange monoliths, one for each substrate addition treatment, water regulation (c) 4 blue monoliths, for water storage and purification measurements and for supporting plant biodiversity (d) 16 coloured monoliths, each colour represents an indicator plant for which direct and indirect plant-soil feedback is estimated phase 2 (P2) on each of four soils conditioned during phase 1 (P1). The monoliths are incubated for 90 days under standard incubation conditions (Table 2). As such the measurements target

the capacity of a soil to deliver key soil functions under optimal conditions for plant growth. For both primary production and biodiversity functions plant harvest days are fixed and based on plant dry mass. Likewise, upon substrate addition (t0) gaseous efflux of $CO_2$, $N_2O$ and $CH_4$ are measured on fixed days, with intensive sampling in the first 14 days, and then less frequent sampling until day 90. In addition, microbial C and C in soil fractions (aggregates) is measured after 70 and 90 days. The water regulation measurements can be done independently in this setup and can potentially be shifted in time, but are

now placed at the end of the 90 day period to spread the workload over time. However, infiltration and leaching measurements will be conducted over a fixed time period.



Table 1. Proposed approach to standardized quantification of the multiple functions of soils. The proposed indicators for each (sub-)function are used to fit a LVM that approximates the generalized soil function. RGR = relative growth rate (g g⁻¹ day⁻¹), C/N = carbon/nitrogen ratio,

| Function | Sub-function | Method | Specifications | Challenges | External validation | Citation |
|---|---|---|---|---|---|---|
| Primary production | Soil fertility | Bio-assay with 4 indicator plant species selected across plant-trait space, using two harvest dry mass RGRs are determined | 4 bio-assay species: - *Festuca rubra* - *Trifolium pratense* - *Arabidopsis thaliana* - *Triticum aestivum* Up to 50 days. | Current species all high light, salt intolerant species. | Biomass production using ingrowth cores in the field | (Daou and Shipley, 2019, 2020) |
| Climate regulation | Carbon stabilization | Soil incubation with 4 substates that differ in C/N. Measure: - Respiration - Microbial biomass C by chloroform fumigation-extraction (after 70 days) | 4 substrates: - Sawdust (C/N >100) - Legume (common bean, C/N ~20-25). - Farmyard manure (C/N ~30-40) - Control | Standardization of substrate quality | - C content in bulk soil and aggregate fractions - Microbial biomass C by chloroform fumigation-extraction | (Doetterl et al., 2015; Laub et al., 2022; Vance et al, 1987) This study |



| Function | Indicator | Method | Timing | Status | Reference |
|---|---|---|---|---|---|
| Water regulation | | - C content in bulk soil and water stable aggregate fractions (after 90 days) | Gas exchange, measure t0, 2-3, 4-6, 14 days intensively, then to | | GHG emissions in situ (Gentile et al., 2008) This study |
| | GHG emission reduction | Measure $N_2O$, $CH_4$ - Indicators are fluxes in the four substates treatments | 90 days less frequently | | |
| | Water storage | - Water infiltration - Water-retention curves using suction cups - Water repellency using water drop penetration time (WDPT) method | - Add fixed volume of water in cylinder on top of soil, measure time to infiltration. - Add water to saturation and lower moisture content using suction cups. - place drops on soil surface and measure time to penetration. | Well established Field based water content | (Doerr et al., 2000) This study |




| | | | | | |
|---|---|---|---|---|---|
| Water purification | - Leachate collected after induced leaching event<br>- Measure contaminant quantity in chemical lab<br>- Optional: Measure ecotoxicity of leachate (and soil). | 4 treatments:<br>- Nutrients: $NO_3$ + $PO_4$<br>- Heavy metals: Cd + Pb<br>- Pesticides: Glyphosate + Fluopyram | Safe laboratory procedures for personnel and safe disposal of toxic waste | Field based lysimeter experiment | (Enell et al., 2016; Lehmann et al., 2020; Schulte et al., 2014)<br>This study |
| Biodiversity | Plant diversity | Phase 1: relative abundance (contribution to evenness)<br>Phase 2: bipartite $I_s$ coefficient & dominant eigenvalue among all the species | Two times 45d | Is four plant species sufficient or need ~30 species? | Measure PSF in in-growth cores and observe biodivesity | (Bever, 2003; Mack et al., 2019; Mack and Bever, 2014)<br>This study |





Table 2. Proposal for standardized incubation conditions and mesocosm setup.

| Factor | Settings |
|---|---|
| Light | 16:8 h day:night, 225 µmol light quanta m$^{-2}$ s$^{-1}$ at plant level |
| Temperature | 26.5° ±2°C (mean ±SD) |
| Air relative humidity | 31% ± 8%. |
| Watering | Add 20 mL water 3 times per week; Monday, Wednesday, Friday. |
| Soil corer | Gouge augur, 60 mm diameter, >25 cm long |
| Container | PVC tube, diameter 60 mm x 25 cm deep (707 cm3) |
| Containers per soil | 32 soil cores = containers |

### 4.1 Primary production

For primary production, I follow the method developed by Daou and Shipley (2019), where they assessed generalized soil fertility. They used four plant species as standard bio-assay indicators that span a wide range in ecological life history strategies (Table S1). Using intact soil cores incubated under fixed environmental conditions in a growth cabinet they estimated the relative growth rates (RGR) of each of the species on each soil. They used that information to fit a measurement model, a specific type of latent variable model, which estimates the values of the latent variable generalized

soil fertility ($F_G$). The measurement model can be thought of as a kind of principal components analysis, but with more constraints imposed on the solution, e.g. that there is one common axis that all four indicator species map onto. They have applied their method successfully to Canadian and French soils with herbaceous plant communities (Daou et al., 2021; Daou and Shipley, 2019), showing that their $F_G$ metric outperforms other metrics as predictors of primary plant production in mixed communities. With my student Judith Nugteren, we applied their method to Dutch grassland soils and our analysis

confirms key aspects of their method (see Supplementary Information). We found that, soils expect to be more fertile based on prior knowledge score higher on the generalized fertility index ($F_G$) and the scores are on the same numerical scale as those of Daou and Shipley (2019), the fertility score is sensitive to fertilizer treatments (Hoagland solution), and replicate soil samples give similar scores indicating a good level of repeatability.





To be representative of generalized soil fertility, and thus primary productivity, the indicator species have to be as ecologically different as possible in order to capture the maximum diversity in responses while being able to grow them together in the same abiotic conditions (light, temperature, soil water levels). Daou and Shipley used herbaceous species of open grassland habitats and chose phytometer species that were (1) as different as possible according to their ecology and taxonomy, (2) have seeds that are easy to acquire by researchers worldwide, and (3) have seeds from recognizable,

reproducible, and stabilized varieties. The selected species (Table S1), cover an interesting gradient of plants, with different root-associated mutualists, growth rates and life span. However, all of them require high light, are salt intolerant, and they do not reflect extreme soil acidities (Lamontagne and Shipley, 2022). The question is thus if indeed these four species are the optimal ones to select when used in an integrated assessment of soil multifunctionality aiming to be applied worldwide?

### 345 **4.2 Climate regulation**

Climate regulation as a soil function has to be split into two sub-functions (Table 1) due to the large differences in soil processes involved: on the one hand carbon storage and on the other preventing emissions of other greenhouse gases (mainly $N_2O$ and $CH_4$; Van de Broek et al., 2019). Carbon is stabilized long-term in the soil when it is fixed to mineral particle matrix or bound in aggregates by microbes (Cotrufo et al., 2019; Lavallee et al., 2020; Lehmann and Kleber, 2015).

This happens through microbial biochemical transformations of rhizodeposits, litter and microbial necromass (Kou et al., 2023; Sokol et al., 2022). The extend to which this happens depends on physico-chemical quality of substrate inputs and the soil matrix properties (Georgiou et al., 2022). Nitrous oxide emissions mainly result from microbial transformations of fertilizers containing reactive nitrogen (Tian et al., 2020; Van de Broek et al., 2019; Zhou et al., 2017), while methane emissions mainly occur under anaerobic conditions when soils are waterlogged and methanogen activity is high (Dalal and

Allen, 2008; Levy et al., 2012). However, soils can also be sinks of methane and nitrous oxide, through methanotrophy and nitrous oxide consumption (Dutaur and Verchot, 2007; Gatica et al., 2020; Tian et al., 2020).

I think we can estimate both sub-functions using the same incubation setup (Table 1, Fig. 3), where we use substrate additions to elicit soil responses. We can estimate carbon stabilization, and thus storage, capacity by incubating a set of four

standard substrates that vary widely in their biogeochemical quality. High N substrates will also induce $N_2O$ efflux. From low to high quality, I propose to use sawdust (C/N >100), farmyard manure (FYM; C/N ~30-40), common bean (*Phaseolus vulgaris*, C/N ~20-25), and a control where nothing is added (only basal respiration). Upon substrate addition, the soils will be incubated at the same conditions as above (Table 2) and gas efflux will be regularly sampled for ~90 days, with intensive

sampling for the first 14 days. Using a gas chromatograph also suitable for quantifying $CO_2$, $N_2O$ and $CH_4$ all three major greenhouse gases could be monitored simultaneously. Since $CO_2$ efflux may not reflect the longer term C fate, I also propose to measure soil microbial C, using chloroform fumigation-extraction (Vance et al., 1987) and C content of soil fractions (bulk soil, large macroaggregates (LMA, > 2 mm), small macroaggregates (SMA, 2–0.25 mm), microaggregates (MiA, 0.25–0.053 mm), and free particles of the silt and clay fraction (SiCl, < 0.053 mm), not included in aggregates; Laub et al., 2022; Six et al., 2000). Microbial C and C in soil fractions will be determined on samples taken on day 70 and 90 respectively (Laub et al., 2022) and analysed using a CN analyser. For substantial $CH_4$ production to occur anaerobic conditions are needed, so sampling for $CH_4$ efflux will need to be combined with the water storage measurements where soil cores are wetted till saturation.

A key challenge here is how to standardize the substrates. The best way would be to implement a standard protocol to purposely cultivate the needed substrates directly, e.g. grow common bean in potting soil under standard conditions, harvest, dry and apply on a mass-basis. For sawdust and farmyard manure this is less straight forward. Instead of FYM, compost may be an alternative, however, for both nutrient content varies among suppliers. So here a global supplier with a well standardized product needs to be identified.

**4.3 Water regulation**

Water regulation has been defined as "the capacity of the soil to remove harmful compounds and the capacity of the soil to receive, store and conduct water for subsequent use and to prevent droughts, flooding and erosion" (Wall et al., 2020). Water storage is the result of a balance between infiltration and runoff during precipitation events, holding water in the soil matrix, and losses to evapotranspiration and percolation to deeper soil layers and aquifers. Water purification is concentrated on the breakdown and sequestration of harmful compounds (Keesstra et al., 2012; Wall et al., 2020).

For water storage capacity I propose to measure infiltration rate, water repellence (hydrophobicity), and to estimate the water retention curve, including water holding capacity. Infiltration is the key input for water in most systems, but lack of infiltration may also importantly impact soil functioning by generating horizontal soil runoff and erosion, and alternatively by waterlogging. To capture these elements a substantial water influx needs to be tested. Water repellence can easily be tested using the water drop penetration time (WDPT) method (Doerr et al., 2000), and reflects an important soil property when they are extremely dry or upon burning, preventing infiltration (Stoof et al., 2011). The water retention curve can be



estimated using standard protocols, see e.g. ISO 11274:2019 (https://www.iso.org/standard/68256.html), e.g. estimating
parameters of the non-linear van Genuchten model. Based on the retention curves estimated values for field capacity (−33
kPa), and permanent wilting point $P_w$ (-1,500 kPa) will be used in the fitting of a latent variable model for water storage
capacity.

With respect to purification (natural attenuation), the EU Water Framework Directive focuses on nutrients, pesticides and
trace elements for groundwater mediated contamination (European Parliament and the Council, 2006). Following
Lehmann et al. (2020) and Wall et al. (2020), I propose to measure $NO_3$ (Nolan and Stoner, 2000), $NH_4$ and P in the leachate
collected after applying a standardized amount of polluted water the soil core to estimate nutrient retention capacity. The
scale used will be % recovery of introduced amount of each nutrient upon measurement using an AutoAnalyzer using
continuous flow analysis. For purification and retention of pesticides (Froger et al., 2023; Tang and Maggi, 2021) water
polluted with Glyphosate and Fluopyram will be added to the soil cores and concentrations measured in the collected
leachate. Glyphosate (https://sitem.herts.ac.uk/aeru/ppdb/en/Reports/373.htm) is a commonly used herbicide. It is the
most leached pesticide globally (Tang and Maggi, 2021) and dominantly found in a French national survey (Froger et al.,
2023), despite being characterized as relatively immobile and low leachable in soils. It is moderately toxic to earthworms,
fish, crustaceans and birds and is, still, approved for used in the EU. Also its major biodegradation product
aminomethylphosphonic (AMPA) needs to be quantified, as it is also toxic to earthworms. Fluopyram
(https://sitem.herts.ac.uk/aeru/ppdb/en/Reports/1362.htm) is a fungicide, with nematicidal side effects, highly leachable
and moderately toxic to aquatic life and earthworms. It is approved in the EU, and frequently found in France (Froger et al.,
2023). Both pesticides can be quantified using reversed phase high-performance liquid chromatography coupled to a
quadrupole mass spectrometer (HPLC-MS/MS; Froger et al., 2023). To estimate heavy metal retention I propose to measure
Pb and Cd concentrations in leachate collected upon application of standardized polluted water to the soil cores. These two
elements can be used to   predict cation heavy metal behaviour, known to negatively affect soil organisms and plants
(Nagajyoti et al., 2010; de Vries et al., 2007), in general. Both can be estimated using flame atomic absorption
spectrometry (FAAS; America Public Health Association, 2017). The required input concentrations of the pollutants for
sensitive indicator use need to be derived empirically.

While I think the response quantification (the indicators) should best be done by assessment of the chemical concentrations
in the leachate, this can be expensive and unfeasible for less resource rich labs. As an alternative I propose to conduct bio-
assays on aquatic life. For instance, algal growth can be used to quantify responses to nutrient leaching and ecotoxicology
protocols (e.g. using *Daphnia* spp.) can be used to assess the toxic potential of the soil leachate. I think the nutrient leachate

needs to contain all assessed nutrients in combination to avoid specific nutrient limitations for the algae. For the toxicity

tests each compound (heavy metal, pesticide) needs to be tested separately to estimate their pure impact. However, it is

known that mixtures are most toxic for soil biodiversity (Beaumelle et al., 2023) and so a treatment where aliquots of each

contaminant are mixed may be critical for extrapolation to field conditions. Furthermore, how direct chemical quantification

and ecotoxicology tests need to be compared across studies requires further study. Likewise, it is an open question whether

responses to such different chemicals can be captured effectively by a single latent variable. Luckily, measurement model

evaluation procedures will quickly inform the researcher if a further division into sub functions is needed.

The impacts of leached contaminants also depends on the subsoil characteristics (Brookfield et al., 2021), so the topsoil flux

estimated here does not inform on the whole impact of a soil on its aqueous surroundings. Indeed, models are needed that

predict the fate of such leached contaminants in a given soil and landscape. Luckily, subsoils are primarily governed by abiotic

properties and processes, less so by biological processes, and modelling could thus be more straightforward.

**4.4 Supporting biodiversity**

For biodiversity, I focus on a soil's potential for supporting plant diversity. Plant diversity within a given location, on the scale

of the interacting plants (Casper et al., 2003), is maintained by preventing or delaying competitive exclusion (Fukami and

Nakajima, 2013; Hardin, 1960). In most terrestrial communities this is importantly mediated by soil-borne antagonists (Bever

et al., 2015; Mack et al., 2019), the net effects of which can be quantified by measuring the soil's plant-soil feedback (Bever,

2003; Van der Putten et al., 2013).

Plant-soil feedback (PSF) is typically measured using a two-phase greenhouse experiment. In the first phase plants are grown

to condition the soil, i.e. they change the soil community and abiotic conditions in their species-specific way (Van der Putten

et al., 2013). In particular they increase the abundance of their associated soil-borne antagonists and mutualists. In the

second or feedback phase, individuals from the same species or a different species are grown in the soil and the difference

in biomass they produce across differently conditioned soils provides information on net plant-soil feedback. Such data can

be used to predict long term coexistence of species using relatively simple mathematical models, that have recently been

extended from pairwise to multi-species models (Bever et al., 1997; Mack et al., 2019). These models can be parametrized

by measuring PSF in a full-factorial soil conditioning and feedback design. Here, I propose to implement such a design for an

artificial community of four plant species, having two growth phases of 45 days each (Fig. 3). From the model we can estimate

the net pairwise interaction coefficient (Is) among the species pairs, but also the real part of the dominant eigenvalue among





all the species, which is a predictive measure for coexistence and stability in the face of local species extinctions (Mack et al., 2019).

**4.5 A new measurement framework for soil multifunctionality**

Once the selected indicators of the multiple soil functions have been measured under standardized conditions for a range of
soils, we can start evaluating the adequacy of the latent variable model for each function. Measurement models for the soil function latent variables can be fit using standard tools used in the social sciences under the term factor analysis (Grace, 2006; Shipley, 2016), this includes ML-based estimation in R package *lavaan* (Rosseel, 2012). Model fit should first be assessed for the component measurement models.

One of the key steps to ensure comparability across labs will be to use internal benchmarks. Benchmarks are used for temperature for instance by fixing the high and low end of the scale to the boiling and freezing point of water, respectively. We can do the same for soil functioning. For instance, for primary production I propose to use pure bare sand (e.g. standard sand used for testing cement; ISO 679:2009(en); https://www.iso.org/standard/45568.html) for the low end of the scale, while high quality potting soil (growing medium) can be used for the high end of the scale (Fig. 4). I predict that also the
subfunctions water storage and purification and carbon storage capacity will be meaningfully mapped using these two internal benchmarks. Whether biodiversity regulation also maps to these two extremes needs to be explored.



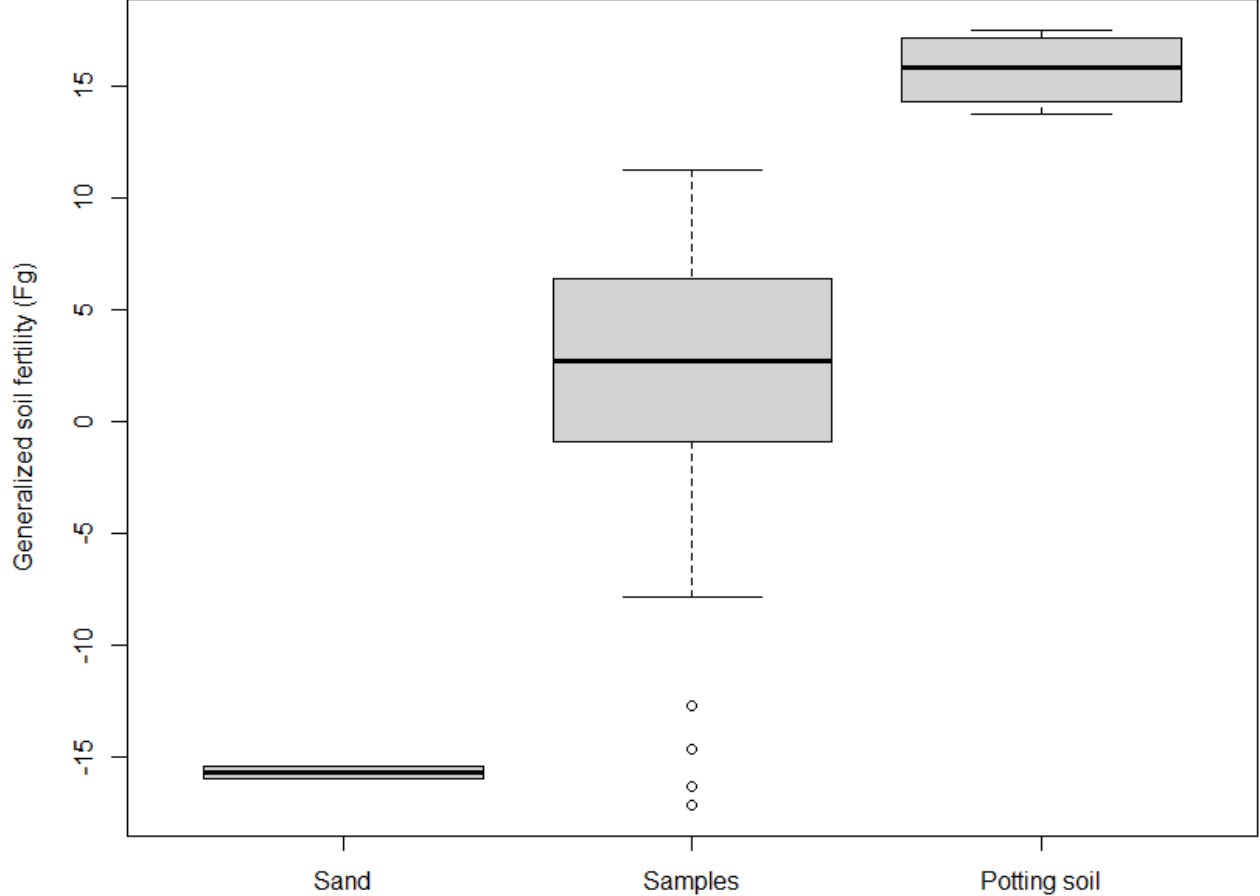

**Figure 4. Using internal controls to benchmark the estimated latent variables representing soil functions.**

Here potting soil and poor sand was used to benchmark the high and low end of the generalized soil fertility scale
respectively. The samples included 30 soils selected from within the Netherlands with contrasting fertility. Soil samples were
taken as field homogenates and incubated in a greenhouse for 50 days. Unless explicitly stated otherwise my student and I
followed the procedures of Daou & Shipley (2019). Four indicator plant species were grown in separate pots for each soil
and harvested, dried and weighed at two points in time per plant species. From these data the relative growth rates per
species and soil was estimated and used to fit a measurement model from which a single latent variable was extracted, called
generalized soil fertility ($F_G$). See Supplementary Information for a detailed protocol, results and discussion.



Another key step will be external validation of the proposed soil function measurement instruments. For this we can leverage long-term established field experiments and research networks such as the ILTER sites for arable systems (Trajanov et al., 2019) and the Nutrient Network for (semi-)natural grasslands (Borer et al., 2017). Within these networks important soil functions are measured, often over time, and provide a good context for comparing the *ex-situ* soil function assessments, quantifying soil potential for soil functioning, proposed here with actual *in situ* measurements. I wonder to what extend to same approaches as I work out here (Section 4) can be used to assess *in-situ* soil functioning as well. Primary productivity and biodiversity regulation can be tested in the field by using in-growth cores in the field directly or using camera systems (rhizotrons; Downie et al., 2015). Lysimeters can be installed to assess leachate contaminations and GHG emissions can be measured in response to substrate additions. This would allow for an explicit 1:1 linkage with the *ex-situ* soil potential measurements, allowing for cross-global comparability, and *in-situ* measurements that estimate real world soil performance. This crucial step can help us build up the causal machinery to link soil intrinsic and extrinsic factors together in a common model to explain and predict soil multifunctionality and thus soil health in reality.

With this proposal to measure the four key soil functions in hand we can put the assessment of soil multifunctionality on a common foundation. Naturally, this is an initial proposal and through discussion and collaboration I think it will need to be refined (see Section 5 for several key concerns and points of improvement). In Fig. 3, there is a schematic representation of the experimental setup needed to implement the proposed scheme. The whole process involves taking 32 soil cores per target soil and incubating them together for 90 days and taking various samples and measurements in the meantime. The setup replies on simple equipment as much as possible. However, critical infrastructure is the incubation facility, e.g. controlled growth cabinets or greenhouse. In addition, a gas chromatograph, CN analyser, AutoAnalyzer, HPLC and FAAS are needed. For labs without access to this high end equipment collaborations with larger labs need to be setup to conduct these analyses. For pollutants, using ecotoxicology approaches represents a low cost alternative, but that needs to be calibrated to the analytical chemistry data. It will be clear that the setup is not feasible for regular soil testing for commercial services, given the long-term incubation period, but that is not the intent. I aim to make a scientific leap forward and for that we need to do the arduous detailed work that we are good at.

## 5 Practical implementation in science and beyond

Generally, the practical and logistical choices for method selection in soil quality assessments varies depending on the objectives: mechanistic understanding, functional land management, and large spatial scale monitoring (Creamer et al., 2022). However, the scheme I propose here aims to strike a balance between these three objectives. The use of



measurement models linked to functions important to land management, standardized measurements that can be compared across labs and thus potentially scaled up, and a flexible framework that allows the integrated study of underlying

mechanisms, make this three-way integration possible. The question is, how well will it do all three?

Currently, I propose that samples be collected as intact soil cores to preserve soil structure and macroscopic features of soil so the real vertical and horizontal variation are reflected in the measurements. However, it was shown that intact cores and homogenized soils generate almost identical pictures of soil fertility (Daou and Shipley, 2020), which would make for much

easier sample collection and handling. Similarly, earlier studies using substrate additions sometimes incubate as little as 80 g of soil (Doetterl et al., 2015), this would strongly minimize substrate and soil requirements and may be an improvement over what I propose here. Likewise, Daou and Shipley (2019) conduct their work in a highly controlled growth cabinet, but could the data still be measured with acceptable error variances in a glasshouse, a screenhouse or in a common garden setup? In a common garden, of course, temperature and light cannot be controlled, but maybe their impact can effectively

be approximated by using growing degree days as measured by a local weather station or temperature loggers? Loosening up this constraint will be important for application of the method in the Global South where high-tech facilities are strongly limiting.

What about sampling time? Do we need to include seasonal dynamics, e.g. reflecting the massive turnover of bacterial and

fungal communities over the year (Schadt et al., 2003), or can we select a single most predictive period? I think it would be most valuable if we could sample in the seasonally cold and/or dry period when plant growth is most limited. Then we could compare *in-situ* soil functioning data in the field during the subsequent growing season to our prior off-season *ex-situ* estimates. These linkages could be used to build predictive models. An alternative would be to sample at peak season, but then often 1) farmers are busy on their field, 2) crops are damaged by sampling and walking, and 3) researchers are occupied

with other field experiments and observations. Also, is it necessary to include an acclimation period where the soil samples are stored prior to experimentation? I propose a 10 day period of the soil cores resting at incubation temperature.

Here, I propose to incubate soils under standard soil-external conditions optimal for plant growth (see Table 2), but can these conditions be applied to all soils? What about soils that experience regular waterlogging? What about soils from low- or high

temperature conditions, will the shift to mesic conditions cause unnatural behaviour of these soils? Can we shorten the protocol? For biodiversity regulation, I propose to conduct two-phase plant-soil feedback experiments (Bever, 1994; Van der Putten et al., 2013), but from the first phase alone we can also use the shoot biomass data to get an initial idea of the soil's



ability to support plant diversity, by looking at the evenness of the relative abundances (Pielou, 1966). Could that be predictive of phase 2 competitive hierarchies?


I am strongly in favour of reporting on the measured soil functions separately so that fellow scientists, policy makers, and the public can make their own assessment and overlay their own priorities with respect to the multiple functions of soil. However, can these measure not be combined in a single indicator? If they are combined with reports of the individual functions I think they can. There is a huge literature on multi-objective optimization methods (Pereira et al., 2022) where

combining objectives is operationalized using explicit rules and criteria. Such optimization should be done with maximum transparency about how functions are weighted and combined for the aggregate index to have any practical use. Also, the weighing should be informed by involving multiple stakeholder group consultation, e.g. using focus group discussions (Bampa et al., 2019; Schulte et al., 2019).

The methods I propose are too cumbersome to be used in commercial soil testing, but crucial to advance our foundational understanding. In order to be useful, indicators need to be conceptually relevant, sensitive to changes, informative for management and effective, e.g. cheap and fast (Lehmann et al., 2020). I argue that my method is conceptually relevant and sensitive and when the measurements are explicitly linked to environmental and management data the results can be used to inform management decisions. The effectiveness is something requiring further testing, see the discussion for steps I want

to take. Additionally we should explore how these soil functioning measurement can be approximated by high throughput screening techniques such as near-infrared spectroscopy, X-ray fluorescence, and potentially eco-acoustics and environmental DNA.

Finally, to scale up and inform spatial planning and management choices worldwide the measurements need to be integrated

in a strong framework, explaining the potential, the synergies and trade-offs among functions mechanistically (Fierer et al., 2021). Including biology in these models is key (Creamer et al., 2022; Fierer et al., 2021). As recent as 2004, a map of known soil threats and degradation published by Science listed only physical and chemical forms of soil degradation and was solely focused on agricultural production (AAAS, 2004). We have moved on, but into unknown territory. The mechanistic machinery is for an important part there in the literature and I am working on a model using plant-microbe-soil stoichiometry

as an organizing principle, but that is too complex for me to present in one paper.





## 6 Discussion

In the wake of the Green Revolution, seeing widespread application of chemical fertilizers and pesticide control, the importance of soil science dwindled. Now, due to the threats exerted on human societies by climate change and biodiversity loss, soil has been revalued as a central nexus integrating many aspects of human wellbeing (Sigl et al., 2023). I believe that

the study of soil multifunctionality and thus soil health should lie at the heart of this new valuation of soil and soil biodiversity, and should be a key focus area in order to bring humanity within the planetary boundaries (Steffen et al., 2015), while simultaneously developing sustainable livelihoods for all (Dearing et al., 2014; Fanning et al., 2022). That also means that we have to put the study of soil multifunctionality on solid empirical and theoretical footing, for which this paper develops a concrete proposal (Section 4; Fig. 3).


A key improvement is that I separated causes and consequences of the soil functions. Focus on the consequences allows standardized measurements that can be adopted across laboratories, both foundational and applied research oriented, and allows them to be linked flexibly, via the estimated latent variables, to competing mechanistic frameworks through structural equations models. Linking the *ex-situ* functional measurements by mechanistic causal models is important also to understand

the results within their environmental context. It is well known that soil health indicators need to be interpreted in site-specific ways (Creamer et al., 2022; Vogel et al., 2018), and that means that a global understanding needs to account for the relevant site-specificities. For instance, clay content determines what range of values to expect for organic matter content (Lehmann et al., 2020), while soil texture shapes ecosystem recovery trajectories (Bach et al., 2010). A key question will be 'how unique are the properties and functions in this soil' compared to the soils in our reference set. To what extend can we

extrapolate our results meaningfully, and based on which (minimum) set of parameters? To answer these questions we need to bring soil functional and contextual measurements together in a common global database.

### 6.1 Outlook

There is a strong need to adjust our spatial planning of land use to best fit to the natural capabilities of soils, for which we

need to know which soils do what functions best. In addition, for optimal management we need to know which functions can be combined for any given soil, and at what level of performance. When both of these aspects are combined we can perform spatial optimization where the service delivery capacity of our soils is explicitly linked to the service provision required by society, e.g. under different climate and socio-economic scenarios (Pereira et al., 2010). In this way we can also get beyond the challenge of different valuation of functions by individual stakeholders (Allan et al., 2015; Lehmann et al.,

2020; Manning et al., 2018), by organizing around societal needs in aggregate.





Here I limited the soil function set to the four key functions from the land management framework (Debeljak et al., 2019; Schulte et al., 2014; Zwetsloot et al., 2021), however soils are involved in more functions, so should we expand the set? What about the quality of the plants produced, we could measure tissue N and protein content, to indicate food and feed quality. What about direct and indirect contributions to human health (Sun et al., 2023; Wall et al., 2015)? Can the soil suppress zoonoses and human disease agents? Does a well-managed soil strengthen the human-associated microbiome and immune systems? Does it reduce allergies? Is it a better source of therapeutics (Thiele-Bruhn, 2021)? What about crop-associated disease suppression (Sagova-Mareckova et al., 2022). For some extend this will be reflected in the primary production and biodiversity functions, but disease agents are often host specific. How can we generate a general picture of the general and specific disease suppressiveness of a given soil? Is that only through sequencing, or can bio-assays of representative pathogens reflect the activity of broad suites of organisms? How will we capture erodibility? This can be done using simulated rain on a standard slope, but how big a surface area do we need minimally? And habitat for soil life or the larval stages of aboveground arthropods? Can we find four indicator species to derive simple tests such as for plant diversity? Do we need eDNA sequencing to predict belowground diversity and composition? What about the predictive capabilities of these measurements? How quickly does their predictive capacity decline over time? Days, weeks, years? What about resistance and resilience to disturbance, should experimental treatments be included in the setup (Harris et al., 2022)? I suppose an additional period of tier 2 testing can easily be implemented once the main measurements have been taken. In short, I have more questions than answers.

**6.2 Conclusions**

Here, I have worked out a simple but causally consistent methodology to consistently quantify soil multifunctionality and thus soil health. The system is based on latent variable modelling (LVM), with each LVM capturing one crucial soil function; primary production, climate regulation (split in carbon storage and GHG emission reduction), water regulation (split in water storage and purification capacity) and biodiversity regulation (captured as plant diversity potential). This system makes explicit that soil functions are complex soil properties, contingent on many drivers, that cannot be measured directly using any device. It also explicitly separates the causes and consequences of each soil function. Using the consequences as indicators we can estimate the LVM factors that approximate the soil intrinsic capacity to perform each function. For example, we can estimate soil fertility from plant growth. I hope this can be a common point of departure in the soil health field to band together and organize the soil multifunctionality and soil health research more mechanistically. In order to optimize sustainable soil health there are two key questions; which soils perform which functions best and what does the synergy-trade-off surface among each function look like across environmental and management gradients? To answer these

questions we need to tackle two challenges: 1) create a common measurement system and 2) come up with a causal model to interlink the soil functions so we can understand and predict their interdependence. These are important steps for us to manage our soils so as to stay within the planetary boundaries (Steffen et al., 2015) and provide sustainable livelihoods for

all (Dearing et al., 2014; Fanning et al., 2022). In this paper, I have proposed a system to tackle challenge 1. I hope you will join me in developing this system for common use.

**Code availability**

R code to fit the soil multifunctionality measurement models and to analyse the Dutch generalized soil fertility model are

available on GitHub: https://github.com/JasperWubs/SoilMFv0.1. This also includes code simulating Simpson's paradox.

**Data availability**

The data for the generalized soil fertility test in Dutch soils is available as Supplementary Data S1.

**Author contribution**

ERJW developed the concept from earlier work of Laurent Daou and Bill Shipley. He worked out the measurement framework

and lead the Dutch generalized soil fertility index experiment and analysed the data. ERJW wrote the paper.

**Competing interests**

I declare to have no conflict of interest.

**Acknowledgements**

I dedicate this paper to Sewall Wright FRS for inventing path analysis and the difficulties he experienced to have his method

accepted. This paper is the result of several years of work on project proposals (eventually successful) and many interactions with colleagues, for which I am very grateful. In particular, I want to thank Bill Shipley (University of Sherbrooke) for introducing me to the concept of measurement models during his Wageningen structural equations modelling course, this is where I got the idea ! I want to thank Johan Six (ETH Zürich) and Paul Bodelier (NIOO-KNAW) for thoughts on measuring soil carbon storage and GHG emissions from soils. Walter Schenkeveld (WUR), Bert-Jan Groenenberg (WUR) and Michiel



Rutgers (RIVM) helped with discussions on measuring the soil's capacity for purification of pollutants. Ciska Veen, Wim van der Putten and Merlijn Schram (all NIOO-KNAW) provided general reflections on quantifying soil multifunctionality and framing the story, thank you. Judith Nugteren (HAS Green Academy) helped me apply the generalized soil fertility index, and some extensions, to Dutch soils (Fig. 4) – thank you for your enthusiasm and diligent work. Finally, I gratefully thank my partner, Ruth van Werven, and my family for all their efforts to support me during good and bad times.

**Funding**


This research was funded by the European Union (MSCA Postdoctoral Fellowship, MultiSol project, #101066007 to ERJW). Views and opinions expressed are however those of the author only and do not necessarily reflect those of the European Union or the European Research Executive Agency (REA). Neither the European Union nor the granting authority can be held responsible for them. The granting authority had no influence in the content of the work.

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
