# Peer review of "Benchmarking soil multifunctionality"

_EGUsphere, 2024_

## Author Response (AR1)

E.R. Jasper Wubs
Netherlands Institute of Ecology (NIOO-KNAW)
Department of Terrestrial Ecology
P.O. Box 50, 6700 AB Wageningen
The Netherlands

Wageningen, 12 February 2025

Dear Dr Merino-Martín, dear Editor,

Herewith I resubmit the revised version of my manuscript entitled 'Benchmarking soil multifunctionality' for further consideration. I am happy with the positive reviewer comments made and the suggestions for improvement.

In response to the reviewer comments I have added section on working with the method in the Global South and added guidelines for the sampling of homogeneous sets of soil cores. I have added reference to current relevant projects and expanded to references in the outlook to support the questions raised there. I went through the entire manuscript to shorten the text where possible and remove digressions present. I also clarified the language where needed.

For further details please see below my detailed responses to the comments raised.

I hope this constitutes a meaningful revision of the manuscript and we look forward to your decision.

Sincerely,

E.R. Jasper Wubs

Reviewer: 1

Dear author,

I've read with great interest your Ms which offers stimulating thoughts on how to assess soil multifunctionnality and soil health. The topic is definitely timely and worth being considered for publication in SOIL. In particular, I've found interesting the concept approach of using LV.

I've however some major recommendations to improve it.

1. I would recommend to give an example of application of this approach with LV. So far, only one example is given for soil fertility, but the different soils functions mentioned in the Ms should be illustrated similarly

    Response: That is indeed something I am planning to do, however, the idea of this paper was to get the concept out and then to leverage the wisdom of the soil science crowd before I/we start measuring for real. So it should be read more as a perspective paper than a research article.

2. The practicability of the approach will definitely be a challenge especially in the global South. I would recommend to discuss this point more in depth

    Response: Thanks for pointing this out. Having worked in the global south this is on my mind a lot and indeed it can be more prominent in the paper. I have added a section to the effect in the ms (L526-530).

3. In the description of the soil sampling strategy, I've questionned myself on how practically speaking homogeneous soil monoliths will be selected. Which criteria should be taken into account to define homogeneous area : for instance soil type x land-use x climate ?

    Response: In general I would recommend to sample as small a space as possible so it is most likely to be homogeneous, and then to take replicate observations through the field and landscape of interest. For sure soil type, land use and climate need to be the same, but I think a finer level of detail is necessary to make the method worthwhile. I added the guidance on this to the ms (L516-517).

4. More references would be necessary in the outlook section where many questions are formulated but the scientific rationale behind each of them should be more justified

    Response: I have added additional references to the section.

5. A review of the existing approaches to benchmark soil multifunctionnality would be necessary especially because there are several Soil Mission projects on this topic : Soil Health Benchmarks project (https://soilhealthbenchmarks.eu/), SOLO project (https://soils4europe.eu/) just to give 2 examples

    Response: thank you for pointing this out. I was part of SOLO and its mission is not to benchmark soil multifunctionality, rather it tries to set roadmaps for future soil research

needs. The soil health benchmarks project is a good approach building on Creamer et al 2022. I now added this information to the text

6. Finally, I would recommend to shorten the Ms and avoid some digressions

    Response: Thanks, I have gone through the ms to shorten it.

I hope these comments will be helpful to revised the Ms.

Yours sincerely,

Julien Demenois

**Citation**: https://doi.org/10.5194/egusphere-2024-2851-RC1

Referee 2

This is an interesting paper that proposes using structural equation modelling to help define soil health. It makes a strong argument for the approach and introduces the reader to the methodology with clearly worked through illustrations of how the method might work. However, this where I had a problem with the paper, as it never went on to apply the proposed method and illustrate its application with some results and then discuss them. For this reason I cannot recommend the paper for publication in SOIL and suggest that it is rejected.

- Response: Thanks for your viewpoint and comments. Indeed, I wrote the paper as a perspectives paper outlining the idea for the method. I did this because I think a good method to measure and quantify soil health is too complicated and too important to do in one go. Rather I aim to leverage the wisdom of the soil science crowd to improve the method before I/we start measuring in earnest.

I have specific comments which are included to help the author improve the paper:

L39 I don't follow the reasoning behind this statement 'Furthermore, soil biodiversity importantly contributes to climate change adaptation, by storing precipitation in soils (Lal, 2020).

- Response: Soil biodiversity helps stabilize SOM which importantly contributes to maintaining water in soils. This link was indeed not explicit and I remedied that in the current version.

L64. What are we looking for in Creamer et al 2022 and Vogel et al 2018?

- Response: Both references come up with conceptual models for modelling the multiple functions of soils.

L129 I don't think SOIL has boxes. Also I am not sure the photograph of Sewall Wright adds very much.

- Response: thanks for pointing this out, I have removed the photograph.

L178 I get the irony, but not really appropriate in a scientific paper

- Response: thanks, I removed it.

L244 I don't agree. Combining methods does not always lead to more accurate results. Combining a poor method with a good one leads to less accurate results.

- Response: I did not mean it in a general sense, but as in the example. Nevertheless, thanks for pointing this out. I included your reasoning in the ms.

L247 A reference is needed for the IQ test for soils.

- Response: the IQ test for soils is the method that is outlined here for the first time in this paper, so there is no independent reference possible. It is introduced, however, in several places before this point in the text.

L254 . I would avoid literary references that not everybody will get.

- Response: okay, I removed it.

L258 figure 2 needs a significantly bfuller caption explaining the different shapes and colours

- Response: I clarified the shapes and colours betters, but other then that I am not sure what the reviewer is referring to. Is it clear right that the caption extends beyond the first bold face line (L255-263)?

L265 What does the * refer to - presumably the figure caption. Incorporate into the caption rather than adding notes like this.

- Response: this was removed an explained in the caption line.

---

## Author Response (AR2)

Response to editors and reviewers

Editor

Dear Jasper Wubs,
Thank you for addressing the reviewers' comments.
After careful consideration of the feedback from both reviewers, it has been decided that your manuscript would be more appropriately categorized as a "forum article" rather than an "original research article", given that no original data are presented. Additionally, please revise your manuscript to better account for existing studies on this topic, as proposed by the first reviewer. If you agree with this change in the type of manuscript, we can proceed with the revision process after addressing the comments.
Best regards,
Luis Merino-Martí

Dear editor, dear Luis,

Thank you for your consideration and decision, I am happy to have the article be presented as a forum article.

I have addressed the comments of the reviewers as detailed below and added the relevant literature.

Best wishes,

Jasper Wubs

Referee report 1

Dear author,

I commend your efforts to consider the recommendations from the different reviewers. I've appreciated that you shortened the Ms and also deleted the digressions. You will find attached some minor comments to be addressed.

Yours sincerely,

Julien Demenois

- Response: Many thanks for your review and suggestions. I have integrated your comments into the new version of the ms.
* * *
Reviewer 2 report

The author clearly feels that presenting a methodology is enough for a perspective paper and, perhaps not surprisingly, disagrees with my initial review. I note that the other reviewers have a different opinion.

- Response: I see the point you have made and appreciate it. Indeed including data was my intention, but since I took a job at another institution I was not allowed to keep the remaining project funds with me to finish my original plan. Nevertheless, I do think there is significant value in this paper, soil health, and measuring soil health is a complicated issue, indeed I know of a number of EU projects scratching their heads and disagreeing as we speak. I am not saying my proposal is the answer to that search, but I do think it is an interesting point of departure. And maybe, this is what I hope, it is better that it is now discussed and augmented first by the scientific community, before we start operationalizing the measurement setup.

While I do agree that SEM and LVM offer an interesting approach to describing soil health, I remain concerned that the paper does not go far enough and that simply presenting an idea for a method or model is not sufficient. I am assuming this is a forum article, since SOIL does not have a 'perspective ' article option. From the SOIL website:

"Forum articles should stimulate an open debate by presenting new ideas and views of soil as part of the larger Earth system. As such, they must strive to be a point of

departure for future work. Purely speculative contributions are discouraged."

My concern is that this paper is not a point of departure for future work. There are already a number of existing papers which have tried to apply SEM to soil health assessments for example Maaz et al. (2023); Romero et al. (2024). These and other papers should have at least been reviewed and the advantages and disadvantages of the approaches that these have taken pointed out. SOIL's readers need to understand how this paper builds on existing work and how it takes us forward beyond the state-of-art. Given the lack of references to previous use of SEM in soil health work I am struggling to understand this.

- Thanks for this perspective and the literature. I think this is where our views diverge, I do think this work is a point of departure for future work. Proper measurement is one of the fundamental steps in any science and for soil health this is something we cannot yet do, not coherently in any case. To me, this is a core discussion to be had.

        - The Maaz et al paper is very interesting, thank you. Their approach to soil health is significantly different from mine. Their range of functions is not as broad as mine, and they use various indicators that are not soil functions, but rather stocks and environmental conditions (TOC, DOC, HWEC, pH, aggregates, bulk density). I think zooming in as closely as possible on the functioning of soil, is critical for quantifying and understanding soil health. It is about construct validity, and I think the properties and conditions used do not deliver that.

- The Romero et al 2024 paper is not using SEM to quantify soil health, it only uses it to link a SH index to ecosystem properties. I disagree with the approach in this paper on several counts. They use the wrong measurements to quantify aspects of soil health (they use stocks and properties not functions) and they use the wrong technique to come to an integration of soil health (Z-scoring with equal weights). More to the point here: they don't use an LVM approach to quantification so it is really fundamentally different. On top of that, I disagree with how they combine machine learning and SEM – to them SEM is just a correlation tool, they ignore the causal-inference framework that should go with the application.
- I have integrated the Maaz et al paper into the new version of the ms.

*Note. I have no scientific relationship to the authors of these papers
Maaz, T. M., Heck, R. H., Glazer, C. T., Loo, M. K., Zayas, J. R., Krenz, A., Beckstrom, T., Crow, S. E. & Deenik, J. L. 2023. Measuring the immeasurable: A structural equation modeling approach to assessing soil health. Science of the Total Environment, 870, 161900.

Romero, F., Labouyrie, M., Orgiazzi, A., Ballabio, C., Panagos, P., Jones, A., Tedersoo, L., Bahram, M., Guerra, C. A., Eisenhauer, N., Tao, D., Delgado-Baquerizo, M., García-Palacios, P. & van der Heijden, M. G. A. 2024. Soil health is associated with higher primary productivity across Europe. Nature Ecology & Evolution, 8, 1847-1855.

---

## Author Response (AR3)

Deventer, 16 June 2025

Dear editors,

Many thanks for your positive decision and support for this ms. I made the textual changes as requested in the new version of the ms.

I must say I don't fully understand the issue with talking about students in the possessive form? I am perenially late to the party with these often-important wokeisms, but I like to know why one form is to be preferred over another :). The supervisor-supervisee relationship is necessarily a hierarchical one I think, similar to parent-child, where I also talk about 'my' children, even though strictly they are not mine at all, they are their own master and they are only temporarily in my care.

Anyhow, thanks for the review and editorial process, it was a pleasure.

Best wishes,

Jasper Wubs